# A First Computational Frame for Recognizing Heparin-Binding Protein

**DOI:** 10.3390/diagnostics13142465

**Published:** 2023-07-24

**Authors:** Wen Zhu, Shi-Shi Yuan, Jian Li, Cheng-Bing Huang, Hao Lin, Bo Liao

**Affiliations:** 1Key Laboratory of Computational Science and Application of Hainan Province, Haikou 571158, China; syzhuwen@163.com; 2Key Laboratory of Data Science and Intelligence Education, Hainan Normal University, Ministry of Education, Haikou 571158, China; 3School of Mathematics and Statistics, Hainan Normal University, Haikou 571158, China; 4School of Life Science and Technology, University of Electronic Science and Technology of China, Chengdu 611731, China; 202211140611@std.uestc.edu.cn; 5School of Basic Medical Sciences, Chengdu University, Chengdu 610106, China; lijian01@cdu.edu.cn; 6School of Computer Science and Technology, ABa Teachers University, Chengdu 623002, China; 20049607@abtu.edu.cn

**Keywords:** heparin-binding protein, amino acid composition, dipeptide composition, dipeptide deviation from expected mean, composition/transition/distribution, support vector machine

## Abstract

Heparin-binding protein (HBP) is a cationic antibacterial protein derived from multinuclear neutrophils and an important biomarker of infectious diseases. The correct identification of HBP is of great significance to the study of infectious diseases. This work provides the first HBP recognition framework based on machine learning to accurately identify HBP. By using four sequence descriptors, HBP and non-HBP samples were represented by discrete numbers. By inputting these features into a support vector machine (SVM) and random forest (RF) algorithm and comparing the prediction performances of these methods on training data and independent test data, it is found that the SVM-based classifier has the greatest potential to identify HBP. The model could produce an auROC of 0.981 ± 0.028 on training data using 10-fold cross-validation and an overall accuracy of 95.0% on independent test data. As the first model for HBP recognition, it will provide some help for infectious diseases and stimulate further research in related fields.

## 1. Introduction

Heparin-binding protein (HBP), also known as azurocidin or CAP-37, is a cationic antimicrobial protein derived from the granulosa protein of polynuclear neutrophils [1,2,3]. Studies have found that the biosynthetic HBP in neutrophils is rapidly released under bacterial stimulation, leading to increased vascular permeability and edema [4,5], and has a proinflammatory effect on a variety of leukocytes and epithelial cells [6]. Therefore, HBP in plasma can be used as a new diagnostic marker for bacterial skin infection, acute bacterial meningitis, leptospirosis, protozoan parasites, and even some noninfectious diseases [4,7,8,9,10]. Especially for sepsis, a systemic inflammatory response syndrome caused by infection, HBP is an effective early and predictive biomarker [11,12,13]. In fact, it has been found that HBP levels in plasma are elevated in septic patients a few hours before the onset of hypotension or organ dysfunction [14].

The correct recognition of HBP can provide important clues for the study of biomarkers of infectious diseases. Traditional molecular biology methods can provide accurate information to study HBP [15,16]. However, these experiments require a longer cycle, more experimental resources, and more expensive manpower. The continuous accumulation of biological data provides a basis for us to mine potential biological knowledge from these data [17,18,19,20,21,22]. The continuous progress of various data analysis methods and artificial intelligence technology provides a favorable tool for us to obtain knowledge [23]. In fact, machine learning methods have been widely used in the recognition of special functional proteins [20,24,25,26,27,28,29,30,31,32,33], for example, bioluminescent protein [34], hormone-binding protein [35], and transcription factors [36,37,38,39,40]. In these works, several kinds of sequence descriptors, such as amino acid composition (AAC), reduced amino acid composition (RAAC) [41,42,43], pseudo amino acid composition (PseAAC) [20,44], and dipeptide composition (DC) [35], were developed.

Although research on these special functional proteins has been successful, to our knowledge, there is still no computational prediction work for HBP recognition at present because there was a lack of available datasets in the past, and people had previously paid more attention to the research of molecular biology experiments. Thus, it is urgent to develop an efficient prediction model to identify HBP.

This work aims to build a powerful computational model to identify HBP. At first, a reliable benchmark dataset was collected and constructed for training and testing various computational models. Subsequently, four sequence descriptors were adopted to formulate sequence samples. Two kinds of machine learning methods, namely, support vector machine (SVM) [45] and random forest (RF) [46], were selected as classifiers for executing classification. The following sections provide a detailed description of the workflow (Figure 1).

## 2. Materials and Methods

### 2.1. Benchmark Dataset Construction

In biological macromolecular classification and recognition, a reliable benchmark dataset is the foundation for constructing a reliable model [47,48,49,50]. It is well known that the Universal Protein Resource (UniProt) is a comprehensive resource for protein sequence and annotation data [51]. This database provides abundant protein information. Therefore, the raw HBP data were collected from UniProt by using heparin binding (KW-0358) as keyword. In UniProt, there are five forms of evidence for protein existence, that is, evidence at the protein level, evidence at the transcriptional level, evidence from homology, predicted, and uncertain. Obviously, the proteins with evidence provided by the first two have higher reliability, so HBPs with the other three kinds of evidence were excluded. In addition, sequences that have ambiguous residues, such as “B”, “J”, “O”, “U”, “X”, and “Z” were checked and excluded. Finally, a total of 391 HBPs were obtained.

Because protein sequences with high similarity will reduce the scalability of the prediction model, those proteins with high similarity must be removed. In general, 40% or 25% is the commonly used threshold of sequence identity when constructing prediction models of special functional proteins. However, due to the limitation of the number of samples, if such sequence identity threshold is used, the number of samples will not be statistically significant, which will lead to the loss of objectivity of the model. Therefore, to balance the number of samples and sequence identity, 80% was adopted as the threshold of sequence identity. The software that performs redundant sequence removal is CD-HIT [52]. As a result, 183 HBPs were kept as positive samples.

In the prediction of special functional proteins, the selection of negative samples is a very difficult task. If all proteins in UniProt that are not annotated as HBP are selected, the data are huge, and the negative samples are almost 3000 times that of the positive samples, which is extremely unfavorable to the construction of the prediction model. In addition, the functional annotation of many of these proteins is not complete. Some proteins may be HBP, but they have not been identified before, which can also lead to bias in a machine learning model. Therefore, to avoid the above two problems as much as possible, the following steps were carried out to select negative samples. First of all, human DNA-binding proteins were chosen as candidate negative samples, because they differ greatly from positive samples in biological functions, which can avoid the problems mentioned above. To improve the reliability of negative samples, those DNA-binding proteins that have structural information, the existence of evidence at the protein level, and a sequence identity of >80% were selected. Despite such stringent criteria, 559 negative sample sequences were obtained, which is more than positive samples. To balance positive and negative samples, 183 of them were randomly selected as the final negative sample dataset.

Based on such benchmark dataset, 50 positive samples and 50 negative samples were randomly selected as test data, and the remaining as training data (133 positives and 133 negatives), formulated as
(1)$$\left\{\begin{array}{c} S_{train}=S_{train}^{positive}\cup S_{train}^{negtive}\\ S_{test}=S_{test}^{positive}\cup S_{test}^{negtive}\; \end{array}\right.$$

### 2.2. Formulation of Protein Sequences

The method for special function protein recognition based on the machine learning method is to classify the samples according to their characteristics in the benchmark dataset [53,54,55,56,57]. Protein is a sequence composed of 20 amino acids with different lengths. However, machine learning requires that every sample should have the same dimension of features. Therefore, how to transform the protein sequence into a discrete digital vector is a key problem for classification model construction. The feature vector should be able to effectively characterize the basic attributes of these samples without losing the original information [58]. In fact, in the past 30 years, scholars have developed a variety of sequence representation methods. Some of these features have good universality, such as pseudo nucleotide composition and position-specific scoring matrix, which have been successful in many protein prediction problems. Additionally, some features have strong specificity, such as amino acid composition, which is very suitable for the recognition of thermophilic proteins. In any previous work, it has not been pointed out what features are used to characterize HBP samples. Therefore, in order to describe the sequence attributes of HBP from multiple perspectives, the following four descriptors were used to extract the features of protein sequences, described as follows.

#### 2.2.1. Amino Acid Composition (AAC)

Although AAC does not perform well in many prediction problems of predicted proteins, it can be used as a supplement to other features as a basic feature [59]. Therefore, this paper also uses this feature to test its prediction performance for HBP.

For any protein sequence expressed as $P=R_{1}R_{2},\cdots ,R_{L}$, where *L* is the length of the protein and also is the number of residues. $R_{i}(i=1,2,\cdots ,L$) is the residue at the *i*-th position in the sequence. *R* belongs to 1 of 20 amino acids (*R*∈(*A*, *C*, …, *Y*)). Then the AAC is the probability that 20 kinds of amino acids appear in this protein, that is, the number of 20 kinds of amino acids divided by the total number of amino acids in this protein (namely, the sequence length *L*). The following formula was used to express the AAC:(2)$$F(R)=\frac{N_{R}}{\sum_{R}N_{R}}=\frac{N_{R}}{L}$$
where *N_R_* is the total number of amino acids *R* in the given sequence P. Then this protein can be expressed by the feature vector as
(3)$$P={\left[F\left(A\right),F\left(C\right),\ldots ,F\left(Y\right)\right]}_{20}$$
where 20 denotes the dimension of the vector.

#### 2.2.2. Dipeptide Composition (DC)

Studies have found that the linkage between amino acids is not random. A certain amino acid is often followed by another relatively fixed type of amino acid; that is, the arrangement of amino acids in proteins is also a unique feature of proteins. In fact, when people study the information storage of the genome, they also find that the adjacent association of nucleotides is the main way of genetic information storage. Therefore, it can also be speculated that the order of amino acids in protein sequences is one way of protein information storage. The order information of adjacent amino acids, also called DC, in proteins is still an important feature to characterize amino acids, and has been widely used in protein classification.

Since proteins have 20 kinds of normal amino acids, there are 400 kinds of dipeptides [60]. We need to count the numbers of these 400 dipeptides in the protein, and then calculate their frequencies in the whole sequence. The calculation formula is as follows:(4)$$F(R^{f}R^{b})=\frac{N_{R^{f}R^{b}}}{\sum R^{f}R^{b}}=\frac{N_{R^{f}R^{b}}}{L-1}$$
where $N_{R^{f}R^{b}}$ is the total number of the dipeptides $R^{f}R^{b}$ in the given sequence P. *f* and *b* represent the front residue *R* and back residue *R* in the dipeptide $R^{f}R^{b}$. Then the feature vector for this protein can be expressed by
(5)$$P={\left[F\left(AA\right),F\left(AC\right),\ldots ,F\left(YY\right)\right]}_{400}$$
where 400 denotes the dimension of this vector.

#### 2.2.3. Dipeptide Deviation from Expected Mean (DDE)

There are 4 nucleotides and 20 amino acids in the genome. The 3 nucleotides are connected to form a codon to encode amino acids or become stop codons. Since 20 kinds of amino acids are encoded by 61 codons, there is degeneracy; that is, one kind of amino acid is encoded by multiple codons. Therefore, for any protein sequence, the theoretical frequency of dipeptide appearance can be described by the coding degeneracy of codons, which is defined as follows:(6)$$TF(R^{f}R^{b})=\frac{C_{R^{f}}}{C_{N}}\times \frac{C_{R^{b}}}{C_{N}}$$
where $C_{R^{f}}$ and $C_{R^{b}}$ are the numbers of codons that code for the first amino acid residue and the second amino acid residue in the given dipeptide “$R^{f}R^{b}$”. $C_{N}$ is the total number of possible codons after excluding the three stop codons ($C_{N}\;$= 61).

Then the theoretical variance of the dipeptide “$R^{f}R^{b}$” can be defined as
(7)$$TV(R^{f}R^{b})=\frac{TF(R^{f}R^{b})\left[1-TF(R^{f}R^{b})\right]}{L-1}$$

The Z-transform is performed between the observed dipeptide frequency (defined in Equation (4)) and the theoretical dipeptide frequency (defined in Equation (6)) in a sequence, as shown below.
(8)$$DDE(R^{f}R^{b})=\frac{F(R^{f}R^{b})-TF(R^{f}R^{b})}{\sqrt{TV(R^{f}R^{b})}}$$

Equation (8) describes the deviation of the observed dipeptide frequency from the theoretical dipeptide frequency and is thus called DDE. Protein sample vectorization is described as
(9)$$P={\left[DDE\left(AA\right),DDE\left(AC\right),\ldots ,DDE\left(YY\right)\right]}_{400}$$
where 400 denotes the dimension of this vector.

#### 2.2.4. Composition/Transition/Distribution (CTD)

Usually, some fragments in a protein chain will form a specific secondary structure or have some special biological activities. Many attempts have been made to describe these fragment features effectively. Among them, CTD is one of the more effective ways to represent the amino acid distribution patterns of a specific structural or physicochemical property in a protein or peptide sequence. Thus, in this work, it is also used for feature extraction to express protein samples.

Amino acid itself is a chemical molecule with specific physicochemical properties. According to the physicochemical properties of amino acids, the frequency of amino acids in each group of properties for a sample sequence (expressed as C) can be redescribed. Amino acids with certain characteristics may form a fragment, such as a continuous hydrophilic fragment exposed on the protein surface. However, the next few amino acids may have other properties. Therefore, T measures the frequencies of property change of amino acids compared with the immediately adjacent amino acids in the sample sequence. D was proposed to character the distribution patterns of the first 25%, 50%, 75%, and 100% of the sample sequence. Details of features are in the following section.

According to the previous studies, 13 physicochemical properties of amino acids are selected for the next characterization. For each property, these 20 amino acids are divided into three categories, such as, for the secondary structure, they can be divided into helix, strand, and coil. Then, 20 amino acids can be divided into 39 total (13 × 3) groups. The percentage of each group in protein sequence is defined as follows:(10)$$C\left(i,j\right)=\frac{n_{i,j}}{\sum_{i,j}n_{i,j}}=\frac{n_{i,j}}{L}$$
where $n_{i,j}$ is the number of residues in the *i*-th group of the *j*-th physicochemical property. Therefore, this descriptor, also known as CTDC, is used to describe the protein sequence as
(11)$$P={\left[C\left(1,1\right),C\left(1,2\right),\ldots ,C\left(3,13\right)\right]}_{39}$$
where 39 denotes the dimension of this vector.

CTDT represents the transition probability of two adjacent amino acid residues belonging to two different groups, which can be calculated by the following formula:(12)$$T\left(i,j\right)=\frac{n_{R^{f}R^{b}}+n_{R^{b}R^{f}}}{L-1}$$
where $n_{R^{f}R^{b}}$ and $n_{R^{b}R^{f}}$ are the numbers of the dipeptides “$R^{f}R^{b}$” and “$R^{b}R^{f}$”, respectively, while $R^{f}$ and $R^{b}$ are amino acids in the *i*-th group and not. Then, vectorization is used to represent the protein as
(13)$$P={\left[T\left(1,1\right),T\left(1,2\right),\ldots ,T\left(3,13\right)\right]}_{39}$$
where 39 denotes the dimension of this vector.

The relative location in one sequence-represented distribution of residues of given groups can be described by CTDD. Considering that a protein sequence is divided into 5 segments according to percentages of 1%, 25%, 50%, 75%, and 100%, the number of amino acids with the *j*-th physicochemical property in group *i* in each segment can be expressed as
(14)$$n_{i,j}^{p}=\left\lfloor \frac{p}{100}\times n_{i,j}\right\rfloor$$
where *p* is 1, 25, 50, 75, and 100. When $n_{i,j}^{p}$ is less than 1, it is assigned a value of 1. Then, CTDD can be represented as
(15)$$D_{i,j}^{\left(1+inter(\frac{p}{25})\right)}=\frac{loc\left(n_{i,j}^{p}\right)}{L}\times 100$$
where $loc\left(n_{i,j}^{p}\right)$ denotes the location at the sequence that the occurrence number of residues of a given group reaches $n_{i,j}^{p}$. Then, the feature vector of CTDD can be expressed as
(16)$$P={\left[D_{1,1}^{1},D_{1,2}^{1},\ldots ,D_{3,13}^{5}\right]}_{195}$$
where 195 denotes the dimension of this vector.

By combining the three features CTDC, CTDT, and CTDD (Equations (11), (13), and (16)), a protein sample could be formulated as a 39 × (2 + 5) = 273 dimensional vector shown as follows:(17)$$P={\left[C\left(1,1\right),\ldots ,T\left(1,1\right),\ldots ,D_{3,13}^{5}\right]}_{273}$$

It should be pointed out that although the three CTD features are mixed in many protein prediction works, the three features exist independently of each other, so they can also be used independently for protein prediction.

### 2.3. Machine Learning Methods

How to find appropriate decision conditions in the feature space, and then distinguish different types of samples, is the third step in the biological macromolecule recognition problem [61,62,63,64,65,66]. Machine learning methods can provide appropriate classification decision criteria to distinguish different types of samples [67]. They have been widely applied in bioinformatics [68,69,70,71,72,73,74,75,76]. At present, deep learning has become a popular method. However, it requires a lot of computing resources and needs more experience to search parameters. Thus, in this work, two popular algorithms, namely, support vector machine (SVM) and random forest (RF), which are very suitable for small samples, were only considered.

SVM is a typical representative of machine learning suitable for small-sample learning. Its principle involves utilizing the kernel function to transform low-dimensional samples into high-dimensional feature space, and then find the hyperplane that can distinguish samples in high-dimensional space. Since most problems in biology are nonlinear subproblems, radial basis function (RBF) are most commonly used. For a detailed introduction to SVM, please refer to the literature.

RF is an integrated learning method based on a decision tree, which can be regarded as an upgrade of a decision tree. Its principle is to construct multiple decision trees for classification during training, and its output is the category selected by most trees. RF avoids the overfitting of a decision tree when building a model.

### 2.4. Evaluation Indexes

After the model is built, the performance of the model needs to be evaluated. In this study, stratified 10-fold cross-validation without shuffle was performed on training data to fine-tune parameters and test models [77,78,79,80,81,82,83,84]. The grid search method was applied to search for the best parameters of the model in search spaces (Table 1). Additionally, the independent data were utilized to test the final model after the best model was established on the training data.

Tenfold cross-validation and independent set test are evaluation strategies for accessing prediction ability. Models also need specific evaluation indicators to evaluate [85,86,87,88,89,90]. Here, the prediction ability of the model was evaluated by using sensitivity (*Sn*), specificity (*Sp*), overall accuracy (*OA*), Matthews correlation coefficient (*MCC*), and area under the receiver operating characteristic curve (auROC):(18)$$Sn=\frac{TP}{TP+FN}$$
(19)$$Sp=\frac{TN}{TN+FP}$$
(20)$$MCC=\frac{TP\times TN-FP\times FN}{\sqrt{\left(TP+FP\right)\times \left(TN+FN\right)\times \left(TP+FN\right)\times \left(TN+FP\right)}}$$
(21)$$A=\frac{TP+TN}{TP+TN+FP+FN}$$
where *TP* and *TN* are the numbers of correctly predicted HBPs and non-HBPs, respectively. *FP* denotes the number of non-HBPs that were recognized as HBPs, while *FN* denotes the number of HBPs that were identified as non-HBPs. Additionally, the auROC can quantitatively evaluate the performance of the model. Thus, as there are several metrics, the auROC is the first metric to be considered. The greater the auROC, the better the performance of the model is. If there are models that have the same auROCs, *OA* can be the second metric to be considered. To be more precise, *MCC* can be the next metric that represents the performances of models.

## 3. Results

### 3.1. Experiments on Training Data

Four kinds of sequence feature extraction strategies were introduced in the above section. According to their definition, each protein can be described as 20 dimension, 400 dimension, 400 dimension, and 273 dimension vectors, respectively, for AAC, DC, DDE, and CTD. Analysis of variance (ANOVA) was applied for ranking the top 5 features of four feature descriptors (Table 2). AAC, DC, and DDE have a common high *F*-score and *p*-value about amino acid *S* (Serine). These results also reveal the extremely different composition between HBPs and non-HBPs. Additionally, according to the high score features of CTD, the differences between HBPs and non-HBPs are mainly concentrated on solvent access, hydrophobicity, polarity, and second structure properties.

Next, the prediction performance of each kind of feature on training data using SVM and RF was investigated. The ROC curves of 10-fold cross-validation are plotted in Figure 2. From Figure 2a, one may notice that *DDE* could produce a maximum auROC of 0.981 ± 0.028 among the four kinds of features when using SVM as classifier. However, the best feature is CTD for RF, as shown in Figure 2b. The auROC is 0.966 ± 0.033. Especially for RF, DDE is the worst feature, which can only achieve an auROC of 0.949 ± 0.049. For AAC and DC, they could produce similar results no matter what kind of classifier was adopted. Through overall comparison, the results of SVM combined with DDE are better. Therefore, this model has the greatest potential to become the ultimate HBP prediction model.

### 3.2. Experiments on Independent Data

On the training data, the prediction results of two algorithms combined with four features were investigated. To further confirm whether SVM combined with DDE is the best prediction model, eight models were tested on independent data (Table 3). It shows that SVM with DDE also has the best *OA* of 95.0%, with a balanced *Sn* and *Sp* of 96.0% and 94.0%, respectively. Other combinations also have the best OA of 95.0%. However, the final model will be chosen according to the highest auROC.

The eight ROC curves are plotted in Figure 3. For SVM-based models, there are two features, DDE and CTD, both of which obtain the maximum auROC. Different from the SVM-based model, the best feature of RF, still CTD, has not changed in the training data and independent data. Additionally, the best parameters for SVM with DDE are shown in Table 4.

## 4. Further Discussion

Based on the above results on training data and independent data, a very good model for HBP recognition was obtained. However, one must realize that the sequence similarity of the benchmark dataset used in this model is relatively high. Generally speaking, building models based on low-similarity datasets has better robustness and scalability. However, the number of current samples is not enough to support us in generating such data for building a model. Therefore, it is our direction to constantly collect new data to expand the samples of the model.

In this work, four features were adopted to encode samples, and numerous sequence features were developed. However, some relatively simple features were utilized to obtain satisfactory prediction accuracy, demonstrating that HBP sequences have their special characteristics. In addition, feature fusion and feature filtering are also commonly used to improve model accuracy, while not applied in this work since the current features have generated considerable prediction performance. Now, although the data size is small, it is sufficient for small sample learning algorithms such as SVM and RF to build light models with pretty good performances. This is also why this first computational work for HBP recognition comes out. Of course, another possible reason for the absence of computational methods is that researchers always focus on experimental methods and ignore them.

With the data size increasing, feature fusion and feature filtering will be implemented when a single feature cannot fully describe the sample characteristics. If there are enough data, more machine learning algorithms can be considered for comparison. Deep learning is now very popular in bioinformatics. However, this algorithm requires more computational resources and also has certain requirements for sample size and feature dimensions.

The identification HBP in medical plasma must consider that the proteins have to obtain their sequences, since our model was constructed based on sequence features. Once sequences of proteins are obtained, feature extraction and model prediction can be conducted, and the results can be produced within a few seconds. Additionally, this procedure only consumes some computing resources—even on mobile phones—without laboratory resources. Compared with molecular biology methods, for example, enzyme-linked immunosorbent assay [91] is based on antibodies of known HBPs, which takes several hours to complete recognition. Computational methods save time and resources. However, there are some limitations, such as false predictions, and the model cannot predict the affinity of heparin binding, which is essential for medical use. In the future, with the development of computational methods and computing resources, more accurate recognition, more functional prediction, and faster processing speed will be achieved. Then more HBPs from various species can be identified, which aids in the research of infectious disease biomarkers.

HBP is an important biomarker of infectious diseases. The correct identification of HBP is of great significance for the study of infectious diseases. The construction of this model will provide clues for the identification of important biomarkers of infectious diseases and the discovery of potential drug targets. This work also contributes to the wider application of artificial intelligence methods in the field of clinical medicine, especially in the identification of biomarkers.

## 5. Conclusions

Four kinds of sequence features were extracted for HBP, and two machine learning methods, SVM and RF, were evaluated. Eventually, DDE combined with SVM was chosen to construct the final prediction model. The model shows good prediction results on both the training set and independent set. To our knowledge, this is the only HBP recognition model based on machine learning. This model is slightly rough, but it provides pioneering research on the use of artificial intelligence methods to study HBP. It is hoped that a more in-depth and detailed analysis of HBP can be carried out in the future.

## Figures and Tables

**Figure 1 diagnostics-13-02465-f001:**
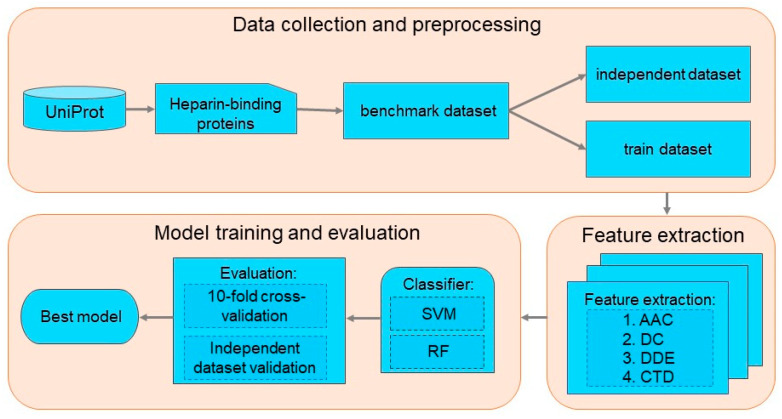
The workflow of the prediction of HBP.

**Figure 2 diagnostics-13-02465-f002:**
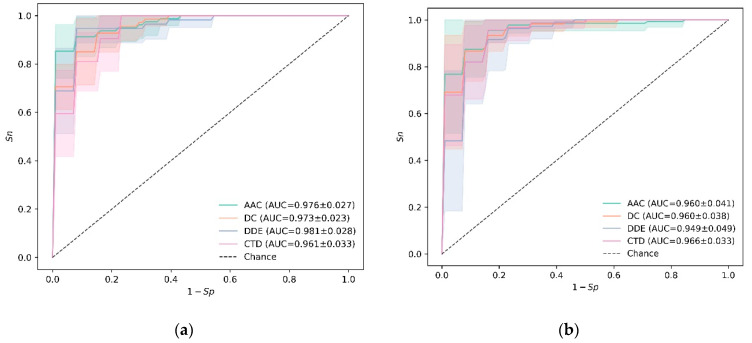
The results on training data using 10-fold cross-validation: (**a**) SVM, (**b**) RF.

**Figure 3 diagnostics-13-02465-f003:**
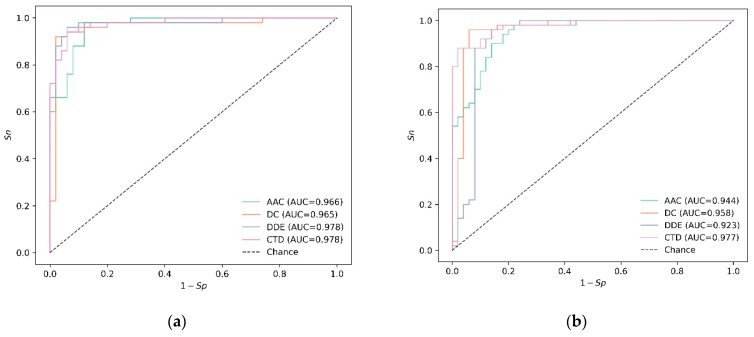
Results on independent data: (**a**) SVM, (**b**) RF.

**Table 1 diagnostics-13-02465-t001:** Search spaces of SVM and RF.

Parameters	SVM ^1^	Parameters	RF
“kernel”	Linear, RBF, sigmoid, poly	“criterion”	Gini, entropy
“C”	2^x^, x ∈ [−1, 15]	“max_depth”	[5, 150]
“gamma”	2^x^, x ∈ [−14, 2]	“min_samples_split”	[2, 30]
“degree”	[1, 5]	“min_samples_leaf”	[5]
\	\	“max_leaf_nodes”	[100]
\	\	“ccp_alpha”	[0.001]
\	\	“n_estimators”	10^x^, x ∈ [1, 3]

^1^ When “kernel” is linear, there are no “gamma” and “degree” parameters to be set. When the only ”kernel” specifies as poly, the “degree” parameter makes sense.

**Table 2 diagnostics-13-02465-t002:** The *F*-scores and corresponding *p*-values for the top 5 features of feature descriptors.

Feature Descriptor	Feature Name	*F*-Score	*p*-Value
AAC	S	105.4221	4.9857 × 10^−21^
	C	51.9136	6.0761 × 10^−12^
	P	39.1761	1.5583 × 10^−9^
	V	28.9828	1.6138 × 10^−7^
	W	18.6945	2.1764 × 10^−5^
DC	SS	85.6575	7.6231 × 10^−18^
	PS	64.2325	3.5827 × 10^−14^
	SP	63.8450	4.1972 × 10^−14^
	PA	39.7520	1.1720 × 10^−9^
	QP	28.6408	4.4865 × 10^−8^
DDE	SS	87.9754	3.1559 × 10^−18^
	PS	62.1955	8.2516 × 10^−14^
	SP	60.4646	1.6840 × 10^−13^
	PA	36.5852	4.9772 × 10^−9^
	FC	29.5543	1.2376 × 10^−7^
CTD	solventaccess.G3	93.4288	4.0577 × 10^−19^
	hydrophobicity_ARGP820101.G2	83.7918	1.5572 × 10^−17^
	polarity.G3	80.3504	5.8770 × 10^−17^
	hydrophobicity_ZIMJ680101.G1	73.3974	8.9800 × 10^−16^
	secondarystruct.G1	69.7518	3.8413 × 10^−15^

**Table 3 diagnostics-13-02465-t003:** Results of models on the independent data using different algorithms and feature descriptors.

Algorithm	Feature	*Sn* (%)	*Sp* (%)	*MCC*	*OA* (%)
SVM	AAC	98.0	88.0	0.864	93.0
	DC	92.0	98.0	0.902	95.0
	DDE	96.0	94.0	0.900	95.0
	CTD	94.0	94.0	0.880	94.0
RF	AAC	90.0	86.0	0.761	88.0
	DC	96.0	94.0	0.900	95.0
	DDE	96.0	86.0	0.824	91.0
	CTD	88.0	98.0	0.864	93.0

**Table 4 diagnostics-13-02465-t004:** Best parameters of SVM with DDE.

Parameters	Value
“kernel”	RBF
“C”	4.59479341998814
“gamma”	0.07982260524725553

## Data Availability

The datasets analyzed in the current study are available in UniProt. Other data are available upon request to the corresponding author.

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
