# Peer review of "A First Computational Frame for Recognizing Heparin-Binding Protein"

_diagnostics, 2023, doi:10.3390/diagnostics13142465_

Round 1

Reviewer 1 Report

Diagnosis-MS

The manuscript entitled (A first computational frame for recognizing heparin-binding protein) by Zhu et al, reported a computational model to identify heparin-binding proteins and its amino acids. The manuscript can be accepted after resolving these issues.

1-    Different typing and grammatic mistakes should be corrected throughout the manuscript. Avoid the use of pronoun ``We``.

2-    The computational study must validate the sequencing of the candidate molecules. Please explain if you ran repeats for the molecular dynamics simulations and provide the full results.

3-    A list of abbreviation should be provided.

4-    Is there is a medicinal purpose of this study. If yes, please clarify it in depth to enrich the discussion part of MS. 

5-    In line 275 the words (400 dimension) repeated two times, correct.

6-      A statistical analysis section should be added to the MS.

7-      If there is a perspective for this protein sequence, if yes please add it in the form of figure.

8-      Concerning to reference part, there are many Journals title abbreviated and others not abbreviated please unify according to the Journal style.

Moderate editing of English language required

Author Response

The manuscript entitled (A first computational frame for recognizing heparin-binding protein) by Zhu et al, reported a computational model to identify heparin-binding proteins and its amino acids. The manuscript can be accepted after resolving these issues.

1-    Different typing and grammatic mistakes should be corrected throughout the manuscript. Avoid the use of pronoun ``We``.

       Response: Thank you very much for your comments. We have made the revision throughout the revised manuscript.

2-    The computational study must validate the sequencing of the candidate molecules. Please explain if you ran repeats for the molecular dynamics simulations and provide the full results.

       Response: Thank you very much for your comment. However, our work is not a study of molecular dynamics. Our work is to use machine learning methods to achieve the recognition of hyperparin binding proteins, and the validation of our model is based on the cross-validation.

3-    A list of abbreviation should be provided.

       Response: Thank you very much for your comments. A list of abbreviations has been added.

4-    Is there is a medicinal purpose of this study. If yes, please clarify it in depth to enrich the discussion part of MS. 

       Response: Thank you very much for your comments. Now more discussions are added in section 4.

5-    In line 275 the words (400 dimension) repeated two times, correct.

       Response: Thank you very much for your comments. There are two 400 dimension vectors. One is for DC, another is for DDE. The current description is correct.

6-      A statistical analysis section should be added to the MS.

Response: Thanks. In this work, we used ANOVA for feature ranking. Thus, we have added the Table 2 in revised manuscript. Please check it.

7-      If there is a perspective for this protein sequence, if yes please add it in the form of figure.

Response: Thank you very much for your comments. It is our understanding that you want to know the statistical analysis of the features of protein sequences. Now, we did ANOVA about different feature extraction methods between positive and negative samples on the training data. However, for a better view, the top five significant features of each feature descriptor were shown in Table 2.

8-      Concerning to reference part, there are many Journals title abbreviated and others not abbreviated please unify according to the Journal style.

       Response: Thank you very much for your comments. Now we have unified styles of journals.

Reviewer 2 Report

The presented work describes a novel development of a machine-learning technique for the identification of Heparin Binding Protein (HBP). By employing Support Vector Machine (SVM) and sequence descriptors, particularly Dipeptide Deviation from Expected Mean (DDE), a robust model for HBP identification was achieved. The study is well-conducted, structured, analyzed, and presented.

However, a point needs further clarification regarding the applicability of the developed technique for HBP identification in plasma, as it is based on sequence descriptors. As presented, it may not seem practical for this specific objective. The authors mentioned that 'traditional molecular biology methods can provide accurate information to study HBP. However, these experiments require a longer cycle, more experimental resources, and more expensive manpower.' A more detailed discussion on this matter could provide better clarity.

The work offers a novel approach to HBP identification using machine learning techniques. It would be beneficial if the authors could provide a more in-depth discussion on how their specific technique based on sequence descriptors can be practically applied for HBP identification in plasma, taking into account the associated limitations and challenges.

Addressing this point would enhance the understanding of their method's usefulness and potential limitations in the clinical context."

Author Response

The presented work describes a novel development of a machine-learning technique for the identification of Heparin Binding Protein (HBP). By employing Support Vector Machine (SVM) and sequence descriptors, particularly Dipeptide Deviation from Expected Mean (DDE), a robust model for HBP identification was achieved. The study is well-conducted, structured, analyzed, and presented.

Response: Thank you for the positive comment.

However, a point needs further clarification regarding the applicability of the developed technique for HBP identification in plasma, as it is based on sequence descriptors. As presented, it may not seem practical for this specific objective. The authors mentioned that 'traditional molecular biology methods can provide accurate information to study HBP. However, these experiments require a longer cycle, more experimental resources, and more expensive manpower.' A more detailed discussion on this matter could provide better clarity.

The work offers a novel approach to HBP identification using machine learning techniques. It would be beneficial if the authors could provide a more in-depth discussion on how their specific technique based on sequence descriptors can be practically applied for HBP identification in plasma, taking into account the associated limitations and challenges.

Response: Thank you very much for your comments. More discussion is added in section 4., please check.

Addressing this point would enhance the understanding of their method's usefulness and potential limitations in the clinical context."

Response: Thanks. We have added some discussion in revised manuscript.

Reviewer 3 Report

Journal: Diagnostics (ISSN 2075-4418)

Manuscript ID: diagnostics-2471554

Title: A first computational frame for recognizing heparin-binding protein

Authors. Wen Zhu, Shi-Shi Yuan, Jian Li, Cheng-Bing Huang*, Hao Lin*, Bo Liao

The authors have carried out a 2x2 classification with two classes heparin-binding protein, HPB and NonHPB. They have applied two machine learning algorithms (support vector machine, SVM and random forest, RF) with five performance parameters (Area under curve of receiver operating characteristic curves AUC; sensitivity, Sn; specificity, Sp; overall accuracy, OA and Matthews correlation coefficient, MCC).

The authors apparently don't know what they were doing. The two modeling algorithms provided different results, and they are also differing in AAC, DDE, DC, CTD, whatever they are. No table(s) is(are) given with performance parameters, the input matrix (vector?) has not been given, either; hence, one cannot check the calculations. In any case the authors have not realized that the performance parameters are conflicting, i.e., only multi attribute decision making is the only suitable choice to achieve any reasonable conclusion.

Recommended reading: Alireza Alinezhad and Javad Khalili: New Methods and Applications in Multiple Attribute Decision Making (MADM) Springer, 2019 (eBook)

https://doi.org/10.1007/978-3-030-15009-9

Some simple geometric concepts (divisions and multiplications) are not entitled to be scientific. In any case the vectorial arrangements are questionable.

The validation is also problematic. Apart from cross-validation many approaches exist. Randomization test, external validation, bootstrap, etc. The way of cross-validation (tenfold) should have been given (blockwise, random, Venetian blind, leave-many-out with and without return, etc.).

I do not see any urgency to elaborate such a framework. Similarly, I do not see the use of it.

I also agree with the authors that “more in-depth and detailed analysis of HBP [should] can be carried out in the future.

Somebody patented two screws at the US patent office as a proof of God’s existence. The authorities checked on novelty, and granted the patent as no earlier such claims were submitted.

Both machine learning techniques produce not one, but endless number of equivalent solutions.

Minor errors

“A first” – indefinite (and definite) articles are not used in titles. The emphasis on primacy only shows the uncertainty of the authors. It is superfluous and not scientific.

“where 20 denotes the dimension of the vector.” – generally a vector is considered one dimensional. Indeed, it can be considered as one point in the n=20-dimensional space. However, a point is zero dimensional.

“Since proteins have 20 kinds of amino acids, there are 400 kinds of dipeptides [59].” – Come on! You probably mean essential (standard) L-amino acids. However, there are D-amino acids, not standard ones, beta-amino acids, and differently substituted ones (e.g., fluorinated, labelled, etc… The number of possible non-natural amino acids is virtually limitless.

“deep learning” – The present example (modeled by SVM and RF) has nothing to do with deep learning or big data.

AUC appeared in figures has not been defined. ROC curves have not been mentioned, at all.

The first sentence in the conclusion is not a concluding remark.

Etc. etc.

In summary the contribution is not ripe for publication. Personally, I doubt that they can complete it to be acceptable. Using freely available programs and clicking on the mouse several times are not scientific.

Author Response

The authors have carried out a 2x2 classification with two classes heparin-binding protein, HPB and NonHPB. They have applied two machine learning algorithms (support vector machine, SVM and random forest, RF) with five performance parameters (Area under curve of receiver operating characteristic curves AUC; sensitivity, Sn; specificity, Sp; overall accuracy, OA and Matthews correlation coefficient, MCC).

The authors apparently don't know what they were doing. The two modeling algorithms provided different results, and they are also differing in AAC, DDE, DC, CTD, whatever they are. No table(s) is(are) given with performance parameters, the input matrix (vector?) has not been given, either; hence, one cannot check the calculations. In any case the authors have not realized that the performance parameters are conflicting, i.e., only multi attribute decision making is the only suitable choice to achieve any reasonable conclusion.

Recommended reading: Alireza Alinezhad and Javad Khalili: New Methods and Applications in Multiple Attribute Decision Making (MADM) Springer, 2019 (eBook)

https://doi.org/10.1007/978-3-030-15009-9

Some simple geometric concepts (divisions and multiplications) are not entitled to be scientific. In any case the vectorial arrangements are questionable.

Response: Sorry for lacking tables for performance metrics and corresponding parameters of classifiers. Now, we added them as Table 3 (Results of models on the independent data using different algorithms and feature descriptors), Table 1 (Search spaces of support vector machine (SVM) and random forest (RF)), and Table 4 (Best parameters of SVM with DDE).

However, when selecting the best model, auROC is our first metric to be considered. The second priority level metric is OA, and then MCC. This priority order is described at the end of section 2.4. The final model that the work suggests is SVM-based model.

The validation is also problematic. Apart from cross-validation many approaches exist. Randomization test, external validation, bootstrap, etc. The way of cross-validation (tenfold) should have been given (blockwise, random, Venetian blind, leave-many-out with and without return, etc.).

Response: Thank you very much for your comments. A more precise description – stratified 10-fold cross-validation without shuffle – has been corrected in section 2.4.

I do not see any urgency to elaborate such a framework. Similarly, I do not see the use of it.

Response: Heparin binding protein (HBP) is a cationic antibacterial protein derived from multinuclear neu-trophils and an important biomarker of infectious diseases. The correct identification of HBP is of great significance to the study of infectious diseases. This work provides the first HBP recog-nition framework based on machine learning to accurately identify HBP.

I also agree with the authors that “more in-depth and detailed analysis of HBP [should] can be carried out in the future.

Response: Thanks.

Somebody patented two screws at the US patent office as a proof of God’s existence. The authorities checked on novelty, and granted the patent as no earlier such claims were submitted.

Both machine learning techniques produce not one, but endless number of equivalent solutions.

Response: We have provided Table 3 in revised manuscript. The final model that the work suggests is SVM-based model.

Minor errors

“A first” – indefinite (and definite) articles are not used in titles. The emphasis on primacy only shows the uncertainty of the authors. It is superfluous and not scientific.

Response: This title is not wrong. Please search papers like “A first/preliminary/novel” or “An initial” with a noun, which tens of thousands of results will be shown there.

“where 20 denotes the dimension of the vector.” – generally a vector is considered one dimensional. Indeed, it can be considered as one point in the n=20-dimensional space. However, a point is zero dimensional.

Response: Thank you very much for your comments. Generally, this kind of description has no problem. The “dimension” of the vector is meaning the “length” of the vector. It is the correct description from mathematics.

“Since proteins have 20 kinds of amino acids, there are 400 kinds of dipeptides [59].” – Come on! You probably mean essential (standard) L-amino acids. However, there are D-amino acids, not standard ones, beta-amino acids, and differently substituted ones (e.g., fluorinated, labelled, etc… The number of possible non-natural amino acids is virtually limitless.

Response: Thank you very much for your comments. We added more precise description of these normal amino acids, excluding ambiguous residues in section 2.1. Moreover, these collected proteins do not contain D-type amino acids.

“deep learning” – The present example (modeled by SVM and RF) has nothing to do with deep learning or big data.

Response: In MS, we have explained that deep learning “requires a lot of computing resources” and sample size is small. “Thus, in this work, we only considered two popular algorithms …” – SVM and RF – “suitable for small samples”. We discussed a point about deep learning here to illustrate that the work here is not suitable for deep learning.

AUC appeared in figures has not been defined. ROC curves have not been mentioned, at all.

Response: Thank you very much for your comments. Now in figures, “AUC” was modified as “auROC”. Descriptions about this metric were added in section 2.4 Evaluation indexes.

The first sentence in the conclusion is not a concluding remark.

Response: Thank you very much for your comments. It was deleted in MS.

Etc. etc.

Response: Thank you very much for your comments. We checked MS and corrected mistakes as we can.

In summary the contribution is not ripe for publication. Personally, I doubt that they can complete it to be acceptable. Using freely available programs and clicking on the mouse several times are not scientific.

Response: We have completed the construction of the entire work model, you don't have to doubt it. In this work, it is not just a click of the mouse that the reviewer imagined, you may have biases here. The entire work requires us to design our own methods and programming in various aspects such as data collection, cleaning, feature extraction, and filtering. I hope the reviewers do not incorporate bias towards calculations into the review process.

Reviewer 4 Report

This study highlights the limitations of traditional molecular biology methods in studying HBP and emphasizes the use of computational methods, particularly machine learning, for HBP recognition. The authors have presented the first HBP recognition framework based on machine learning to accurately identify HBP. However, certain parts of the manuscript are unclear and insufficiently described.

Specific comments are as follows:

1.      The introduction should be clearer and provide a brief explanation of the relevance and significance of computational HBP recognition. Additionally, it would be beneficial to explain how the computational model streamlines the analysis process compared to traditional molecular biology methods. Furthermore, the authors should discuss how the model reduces the time and resources required for HBP analysis, ultimately enabling faster and more cost-effective investigations.

2.      The manuscript mentions the absence of computational prediction work for HBP recognition, but it does not clearly explain why this gap exists. It would be helpful to provide a brief explanation, such as the lack of available datasets or a previous focus on experimental methods.

3.      The study should discuss its limitations, including any potential sources of error or uncertainty in the computational model. Addressing these limitations will enhance the credibility of the research.

4.      Emphasize the novelty of the proposed computational model for HBP recognition based on machine learning. Discuss the potential impact of this model on the field of infectious disease biomarker research and how it contributes to the broader application of artificial intelligence methods.

5.      In addition to suggesting more in-depth and detailed analysis of HBP in the future, provide specific directions for future research. For example, discuss expanding the dataset, exploring other sequence features, incorporating feature fusion and filtering techniques, and considering additional machine learning algorithms for comparison.

By incorporating these changes and additions, the revised manuscript will provide a more comprehensive and informative review of the study on HBP and its computational prediction model.

Minor editing of English language required.

Author Response

This study highlights the limitations of traditional molecular biology methods in studying HBP and emphasizes the use of computational methods, particularly machine learning, for HBP recognition. The authors have presented the first HBP recognition framework based on machine learning to accurately identify HBP. However, certain parts of the manuscript are unclear and insufficiently described.

Specific comments are as follows:

  1. The introduction should be clearer and provide a brief explanation of the relevance and significance of computational HBP recognition. Additionally, it would be beneficial to explain how the computational model streamlines the analysis process compared to traditional molecular biology methods. Furthermore, the authors should discuss how the model reduces the time and resources required for HBP analysis, ultimately enabling faster and more cost-effective investigations.

       Response: Thank you very much for your comments. More explanations are added in section 4.

  1. The manuscript mentions the absence of computational prediction work for HBP recognition, but it does not clearly explain why this gap exists. It would be helpful to provide a brief explanation, such as the lack of available datasets or a previous focus on experimental methods.

Response: Thank you very much for your suggestion. Indeed, there was a lack of available datasets in the past, and people had previously paid more attention to the research of molecular biology experiments.

  1. The study should discuss its limitations, including any potential sources of error or uncertainty in the computational model. Addressing these limitations will enhance the credibility of the research.

Response 4: Thank you very much for your comments. We have added the description in section 4.

  1. Emphasize the novelty of the proposed computational model for HBP recognition based on machine learning. Discuss the potential impact of this model on the field of infectious disease biomarker research and how it contributes to the broader application of artificial intelligence methods.

       Response 4: Thank you very much for your suggestion. Heparin binding protein (HBP) is an important biomarker of infectious diseases. The correct identification of HBP is of great significance for the study of infectious diseases. The construction of this model will provide clues for the identification of important biomarkers of infectious diseases and the discovery of potential drug targets. This work also contributes to the wider application of artificial intelligence methods in the field of clinical medicine, especially in the identification of biomarkers.

  1. In addition to suggesting more in-depth and detailed analysis of HBP in the future, provide specific directions for future research. For example, discuss expanding the dataset, exploring other sequence features, incorporating feature fusion and filtering techniques, and considering additional machine learning algorithms for comparison.

       Response: Thank you very much for your comments. Since most of these points have already been mentioned in the discussion, we just add a point about more machine learning algorithms in the discussion: “If there is enough data, more machine learning algorithms can be considered for comparison”.

By incorporating these changes and additions, the revised manuscript will provide a more comprehensive and informative review of the study on HBP and its computational prediction model.

Response: Thank you very much.

Round 2

Reviewer 1 Report

No comments 

Fine

Reviewer 3 Report

Journal.: Diagnostics

Title: A first computational frame for recognizing heparin-binding protein

Manuscript ID: diagnostics-2471554-peer-review-v2

Authors: Wen Zhu, Shi-Shi Yuan, Jian Li, Cheng-Bing Huang, Hao Lin*, Bo Liao*

The authors have accepted a minimum change approach. My queries remained unanswered mostly: “Some simple geometric concepts (divisions and multiplications) are not entitled to be scientific. In any case the vectorial arrangements are questionable.”

The authors’ response is valid and true:” Sorry for lacking tables for performance metrics and corresponding parameters of classifiers. Now, we added them as Table 3 (Results of models on the independent data using different algorithms and feature descriptors), Table 1 (Search spaces of support vector machine (SVM) and random forest (RF)), and Table 4 (Best parameters of SVM with DDE).” Unfortunately, the response has nothing to do with the query.

I admit Table 3 is new and contains some information. However, my calculations do not justify its usefulness, even worse it does not justify the completion of calculations.

The fair method comparison called some of ranking differences (SRD) suggest equivalency of all methods with one exception. A hypothetical best method was defined as gold standard (i.e., the row maximums:

The general statement that SVM is superior is NOT justified, and almost all methods are indistinguishable from random ranking (blue line is right to the 5% limit). The calculations are reproducible, a link to a downloadable program can be found in ref. [K. Kollar-Hunek, K. Heberger, Method and model comparison by sum of ranking differences in cases of repeated observations (ties) Chemometrics and Intelligent Laboratory Systems, 127 (2013) 139-146. http://dx.doi.org/10.1016/j.chemolab.2013.06.007 ].

Authors: “However, when selecting the best model, auROC is our first metric to be considered.”

Why? There is no justification for that. The performance merits are conflicting; hence, only multicriteria decision making (Parato optimization) is the only feasible choice.

The problematic validation cannot be solved by “ more precise description – stratified 10-fold cross-validation without shuffle – has been corrected in section 2.4.” – One validation technique is NOT sufficient. Not to speak about the contradictory results of the various validation methods.

My opinion has not been changed “I do not see any urgency to elaborate such a framework. Similarly, I do not see the use of it.” I admit the importance of great significance of HPB. However, machine learning algorithms reproduce only the existing knowledge. No prediction is possible.

My example about God existence illustrates the case that alone the novelty is not scientific.

Again, the authors could not respond to my comment. “Both machine learning techniques produce not one, but endless number of equivalent solutions.” [because of the large number of regularization parameters]. You can always select condition when SVM superior or far inferior to RF. Moreover, comparison of JUST two techniques leads nowhere.

Response: We have provided Table 3 in revised manuscript. The final model that the work suggests is SVM-based model.

Unfortunately, the title is wrong. Referring to wrong papers does not help, containing the phrase “A first/preliminary/novel” in tens of thousands of cases means that much garbage is present in the literature.

Modifying “AUC” to “auROC” is inadequate. The other minor errors were misunderstood or not properly solved.

To design your “own methods and programming in various aspects such as data collection, cleaning, feature extraction, and filtering” is a routine activity and becomes scientific if significant conclusions are achieved.

As the authors ignored my suggestions and the Table 3 have not provided sufficient proofs I insist to rejection. I would not even encourage revision and resubmission, or selection another journals.

July 19 / 2023             referee:

(The figure can be found in the attached pdf file)

Reviewer 4 Report

The manuscript has undergone substantial improvements and is now ready for publication.

Minor editing of English language required.